# Nitrogen Doped Porous Biochar/β-CD-MOFs Heterostructures: Bi-Functional Material for Highly Sensitive Electrochemical Detection and Removal of Acetaminophen

**DOI:** 10.3390/molecules28062437

**Published:** 2023-03-07

**Authors:** Qi Yu, Jin Zou, Chenxiao Yu, Guanwei Peng, Guorong Fan, Linyu Wang, Shangxing Chen, Limin Lu, Zongde Wang

**Affiliations:** 1East China Woody Fragrance and Flavor Engineering Research Center of NF&GA, College of Forestry, Jiangxi Agricultural University, Nanchang 330045, China; 2Key Laboratory of Crop Physiology, Ecology and Genetic Breeding, Ministry of Education, Key Laboratory of Chemical Utilization of Plant Resources of Nanchang, College of Chemistry and Materials, Jiangxi Agricultural University, Nanchang 330045, China

**Keywords:** N-SC/β-CD-MOFs composite, heterostructures, electrochemical detection, adsorption, acetaminophen

## Abstract

Acetaminophen (AC) is one of the most common over-the-counter drugs, and its pollutant in groundwater has attracted more attention due to its serious risk to human health. Currently, the research on AC is mainly focused on its detection, but few are concerned about its removal. In this work, for the first time, nitrogen-doped Soulangeana sepals derived biochar/β-cyclodextrin-Metal-organic frameworks (N-SC/β-CD-MOFs) composite was proposed for the simultaneous efficient removal and detection of AC. N-SC/β-CD-MOFs combined the properties of host-guest recognition of β-CD-MOFs and porous structure, high porosity, and large surface area of N-SC. Their synergies endowed N-SC/β-CD-MOFs with a high adsorption capacity toward AC, which was up to 66.43 mg/g. The adsorption type of AC on the surface of N-SC/β-CD-MOFs conformed to the Langmuir adsorption model, and the study of the adsorption mechanism showed that AC adsorption on N-SC was mainly achieved through hydrogen bonding. In addition, the high conductivity, large specific surface area and abundant active sites of N-SC/β-CD-MOFs were of great significance to the high-performance detection of AC. Accordingly, the sensor prepared with N-SC/β-CD-MOFs presented a wide linear range (1.0–30.0 μM) and a low limit of detection of 0.3 nM (S/N = 3). These excellent performances demonstrate that N-SC/β-CD-MOFs could act as an efficient dual-functional material for the detection and removal of AC.

## 1. Introduction

In recent years, pharmaceutical pollutants in the groundwater have aroused great concern because of their great threat to the environment and public health [1]. Acetaminophen (AC) as a common anti-inflammatory drug is widely used to treat fevers, colds and alleviate postoperative inflammation. However, there are two sides to every coin [2]. The widespread use of AC results in large amounts of AC being discharged into groundwater. According to a recent research report [3], the concentration of AC in the freshwater ecosystem has reached up to ng/L or even μg/L, which will cause potential harm to non-target aquatic organisms. Considering that detection alone will not solve the problem of AC pollution in water, an effective strategy for simultaneous detection and removal of AC in water needs to be developed urgently. Up until now, adsorption has been considered a reliable technology for removing pollution from groundwater because of the advantages of excellent removal effect and ease of operation [4]. At the same time, electrochemical detection technology is widely used in various fields due to its high accuracy, rapid detection, easy miniaturization and real-time monitoring [5,6]. Considering that the performance of the electrochemical sensor is primarily decided by two aspects, one being the enrichment ability of electrode modification materials toward the detection substances and the other being their catalytic performance. Therefore, it is feasible to propose a material with both good electrocatalysis and adsorption capacity to achieve simultaneous sensitive detection and efficient removal of pollutants.

β-cyclodextrin (β-CD) is a cyclic oligosaccharide consisting of seven glucose units, which has a unique structure of a hydrophilic outer surface and hydrophobic inner cavity [7]. Those cavities endow them selectively capture some organic molecules through a well-defined “host–guest” recognition mechanism [8]. However, the agglomeration of β-CD would result in a large number of active sites being covered, decreasing poor enrichment performance [9]. To solve this problem, β-CD-metal organic frameworks (β-CD-MOFs) with an ordered porous structure, a large specific surface area and high porosity were proposed, which can avoid the agglomeration of β-CD and fully expose the adsorption sites on the surface of β-CD-MOFs [10]. Based on the above properties, β-CD-MOFs are promising as a good adsorbent for AC removal and an effective enrichment in electrochemical detection. However, it also should be noted that the inherently poor conductivity of MOFs is not conducive to sensing detection [11]. In order to obtain good sensing performance, it is necessary to introduce some materials with good conductivity and catalytic properties to combine with β-CD-MOFs.

Carbon materials derived from biomass have been active in various research fields (especially electrochemical sensing and adsorption) due to their wide sources, cheap, easy availability, easy modification, large specific surface, excellent conductivity, and catalytic properties [12]. It is also reported that doping of N in the carbon skeleton could break the inertness of sp3 hybridized C and create abundant defects and effective active sites, which greatly improves the conductivity and catalytic performance of carbon material [13]. For example, Huang et al. developed an enhanced electrochemical platform based on lotus leaves and derived N,P-doped porous carbon nanoparticles for the determination of ascorbic acid with a low limit of detection (LOD) (4.25 μM) [14]. Zou et al. synthesized nitrogen self-doped porous carbon derived from a cicada shell for electrochemical determination of Cu^2+^ with a relatively low LOD of 25.24 nM [15]. Liu et al. reported that the N-doped porous biochar nanocomposite was used as an electrochemical sensor to simultaneously detect catechol and hydroquinone with LODs of 37 and 47 nM [16]. The ordered pore structure is beneficial to the transfer of electrons and materials on the electrode surface, as well as adsorption [17]. Therefore, it is necessary to conduct controllable pore-forming on the surface of biomass-derived carbon materials. Although the biomass will be self-activated when carbonized at high temperatures, the pore structure obtained by such self-activation is uncontrollable. In contrast, the use of various activators, such as KOH [18], KCl [19], ZnCl_2_ [20], and K_2_CO_3_ [21], could realize the orderly adjustment of porous structure in the preparation of biochar.

Based on the above, N-doped SC (N-SC)/β-CD-MOFs heterostructures were proposed for the detection and removal of AC (Figure 1). The synergistic effect of β-CD-MOFs and N-SC endowed N-SC/β-CD-MOFs with excellent conductivity, catalytic performance, and good adsorption capacity. At the same time, the pore space effect, fast electron mobility, and induced adsorption effect of the heterostructures could further improve the adsorption and catalytic performance of the composite [22], which makes N-SC/β-CD-MOFs heterostructures promising materials for the detection and removal of AC. These results indicated that the N-SC/β-CD-MOFs displayed excellent detection and adsorption performance toward AC. In addition, the electrochemical AC sensor based on N-SC/β-CD-MOFs presented good stability, high sensitivity, and excellent selectivity, and realized the detection of AC in actual water samples successfully.

## 2. Result and Discussion

### 2.1. Material Characterization

The morphological structures of NKSC, N-SC, β-CD-MOFs, and N-SC/β-CD-MOFs were investigated using SEM. From Figure 1A, the NKSC presented a large-scale, blocky shape. Nevertheless, N-SC exhibited a 3D honeycomb-like porous structure (Figure 1B), which might be because KOH activation led to a large amount of gas release from the biochar surface [23]. Figure 1C displayed that β-CD-MOFs was a blocky crystal with a particle size of 5–10 μm, which is consistent with the previous literature [24]. After mixing with N-SC (Figure 1D), β-CD-MOFs were uniformly embedded in porous N-SC. In general, the result of SEM proves that the composite has been successfully synthesized.

The crystalline structures of β-CD-MOFs, N-SC, and N-SC/β-CD-MOFs were investigated by XRD (Figure 2A). The XRD profile of β-CD-MOFs exhibited a series of characteristic peaks between 5° and 30°, which are in good accordance with previous reports [25]. For N-SC, two broad diffraction peaks were observed around 24° and 43°, corresponding to the (002) and (100) lattice planes of amorphous carbon [26]. The XRD of N-SC/β-CD-MOFs retained both the characteristic peaks of N-SC and β-CD-MOFs, revealing that mixing with N-SC did not destroy the structure of β-CD-MOFs [11]. Above all, these results also confirmed that the N-SC/β-CD-MOFs composite was prepared successfully. Besides, the XRD of N-SC/β-CD-MOFs + AC was similar to that of N-SC/β-CD-MOFs, demonstrating that the adsorption of AC did not destroy the crystal structure of N-SC/β-CD-MOFs, which further proved the stability of N-SC/β-CD-MOFs.

The element types on the surface of N-SC/β-CD-MOFs were characterized with XPS. The result showed that there were four main elements (C, O, N, and K) in N-SC/β-CD-MOFs, and their contents were 64.9%, 26.7%, 3.7%, and 4.7%, respectively (Figure 2B), indicating that N was successfully doped. Subsequently, high-resolution spectrums of each element were studied to further analyze the state of the elements. The high-resolution spectrum of C 1s (Figure 2C) showed the superposition of three peaks, which were the C-C (283.81 eV), C-N (284.80 eV), and C=O (286.15 eV), respectively [27]. The peaks at 530.82 and 532.39 eV in the O 1s spectrum could be assigned to C-O and C=O. The N 1s emission spectrum displayed the presence of four doping forms (Figure 2E); pyridinic nitrogen (398.12 eV), pyrrolic nitrogen (398.90 eV), graphitic nitrogen (399.91 eV), and oxidized nitrogen (401.22 eV), respectively [28]. The result indicated that pyrrole-type nitrogen was the main existing form in the N-SC/β-CD-MOFs composite, and pyridine-type nitrogen and pyrrole-type nitrogen were beneficial to increase the active sites, thereby improving the catalytic activity [29]. Besides, two obvious peaks of 293.76 eV and 296.37 eV were found in the high-resolution spectrum of K 2p (Figure 2F), further demonstrating that β-CD-MOFs constructed with β-CD and potassium ions were successfully prepared and the structure was consistent with the results reported previously [11].

### 2.2. Adsorption Studies

#### 2.2.1. Effect of Adsorption Conditions

The pH is a crucial factor in an adsorption study since it affects both the surface charge of the adsorbent and the adsorbate. In Appendix A, N-SC/β-CD-MOFs (0.77 g·L^−1^) were employed to adsorb 65 mg/L AC with different pHs (3.0–11.0) at room temperature for 240 min to explore the optimal pH of an AC solution. The result demonstrated that the largest adsorption capacity (64.47 mg·g^−1^) and removal efficiency (76.30%) appeared when the pH was 7.0. This phenomenon might be attributed to the fact that when the pH was less than 7.0, the surface of N-SC/β-CD-MOFs was positively charged due to the adsorption of a large amount of H^+^. The adsorption capacity gradually increased with the increase in pH from 3.0 to 7.0, which was attributed to the repulsive interaction between the surface of N-SC/β-CD-MOFs and the carbocation of AC. The repulsive interaction decreased with the further increase in pH. However, when pH exceeded 7.0, AC mainly existed in ionized (base) forms because its pKa value was 9.4. The surface of N-SC/β-CD-MOFs was electronegative due to OH^−^, so there was a strong repulsive force between AC and N-SC/β-CD-MOFs [30]. Therefore, 7.0 was used as the optimal pH for the next adsorption studies of AC.

The effect of N-SC/β-CD-MOFs dosage on the adsorption efficiency of AC was explored by adsorbing AC (65 mg·L^−1^) with various concentrations (0.19, 0.77, 1.35, 1.92 and 2.5 g·L^−1^) of N-SC/β-CD-MOFs (Appendix A). The result showed that the removal efficiency increased with the increase in N-SC/β-CD-MOFs dosage in the range of 0.19–1.35 g·L^−1^, which might be owing to the fact that more available active sites were provided for AC with the increase in N-SC/β-CD-MOFs. However, the removal efficiency kept plateauing when the dosage of N-SC/β-CD-MOFs exceeded 1.35 g·L^−1^, which might be due to the aggregation or overlapping of active sites on N-SC/β-CD-MOFs. In addition, the results also showed that the adsorption capacity (q, mg·g^−1^) decreased with the increase in the amount of N-SC/β-CD-MOFs [31]. Therefore, considering both adsorption capacity and removal efficiency, 0.77 g·L^−1^ was chosen as the optimal amount of N-SC/β-CD-MOFs for subsequent adsorption experiments.

#### 2.2.2. Adsorption Kinetics

The adsorption capacity (q, mg·g^−1^) of β-CD-MOFs, N-SC, and N-SC/β-CD-MOFs toward AC were investigated at the optimized pH (Figure 3A). It could be clearly seen that N-SC/β-CD-MOFs showed the highest adsorption capacity, compared with β-CD-MOFs and N-SC. This result demonstrated that the synergistic effect between β-CD-MOFs and N-SC endowed N-SC/β-CD-MOFs with excellent adsorption performance toward AC.

Subsequently, the relationship between adsorption time and adsorption capacity was researched in Figure 3B. It revealed that the adsorption capacity of AC increased rapidly within 0–30 min, and then hold steady, indicating that the adsorption process of N-SC/β-CD-MOFs toward AC was very rapid. The rapid adsorption at 0–30 min can be interpreted as the fact that the abundant binding sites could enhance the contact between the adsorbent and the adsorbate [32]. However, as the adsorption progressed, the adsorption reached equilibrium when all the adsorption sites on the surface were occupied, and then the adsorption capacity would not change after 30 min. To further explore the adsorption process between N-SC/β-CD-MOFs and AC, the experimental data were fitted with the pseudo-first-order (Equation (1)) and pseudo-second-order (Equation (2)) models, respectively [33]. Their fitting plots were shown in Figure 3B, and the relevant adsorption kinetic parameters were summarized in Table 1. The R^2^ value of the pseudo-second-order model (0.9975) was larger than that of the pseudo-first-order model (0.9864), demonstrating that the adsorption process between N-SC/β-CD-MOFs and AC belonged to the pseudo-second-order model. It indicated that the interactions between N-SC/β-CD-MOFs and AC were mainly controlled by chemical processes:
(1)
$$q_{t}=q_{e}\left(1-e^{-k_{1}t}\right)$$


(2)
$$q_{t}={q_{e}}^{2}k_{2}t/\left(1+q_{e}k_{2}t\right)$$

where *q_e_* (mg/g) represents the equilibrium adsorption capacity of AC; *q_t_* (mg/g) was the adsorption capacity at one point; t represented time; and *k*_1_ (min^−1^) and *k*_2_ (g/mg⋅min) were the pseudo-first- and pseudo-second-order rate constants, respectively.

#### 2.2.3. Adsorption Isotherm Analysis

For the first time, the relationship between the initial concentration of AC and adsorption capacity was studied (Figure 4A, black line). The result showed that with the increase in the initial concentration of AC, the adsorption capacity increased from 1.18 to 91.39 mg/g, and the rate of increase slowed down fast. This might be put down to the fact that the concentration difference between the solution and the surface of N-SC/β-CD-MOFs gradually increased with the increase in the initial concentration of AC, leading to the acceleration of the diffusion rate of AC. Thus, the adsorption performance was improved. However, the limited sites on the surface of N-SC/β-CD-MOFs would be almost all occupied when the initial concentration of AC continued to increase, and when the adsorption reached equilibrium [34]. Therefore, the increase in adsorption capacity would slow down as the initial concentration of AC continued to increase. Nevertheless, the removal percentage of AC decreased from 90.54% to 23.43% with increasing the initial concentration of AC (Figure 4A, red line). Then, the research of adsorption isotherm was performed to reveal the adsorption type of AC on N-SC/β-CD-MOFs (Figure 4B). Two kinds of adsorption isotherm models, the Langmuir model and the Freundlich model, were employed to fit the adsorption isotherm of AC. The Langmuir (Equation (3)) and Freundlich (Equation (4)) isotherm equations were expressed as follows [35,36]:
(3)
$$q_{e}=q_{m}c_{e}K_{L}/\left(1+c_{e}K_{L}\right)$$


(4)
$$q_{e}=K_{F}{c_{e}}^{1/n}$$

where *q_e_* (mg/g) is the equilibrium adsorption capacity of AC on N-SC/β-CD-MOFs. *q_m_*(mg/g) is the maximum adsorption capacity of AC on N-SC/β-CD-MOFs. *c_e_* (mg/L) is the equilibrium concentration of AC. *n* (dimensionless) is the Freundlich intensity, *K*_L_ (L/mg) is the Langmuir constant, and *K_F_* (mg/g (L/mg)^1/n^) is the Freundlich constant.

The nonlinear fitting diagram of Langmuir (red line) and Freundlich (blue line) isotherm models with the adsorption isotherm of AC was shown in Figure 4B, and the relevant fitting parameters of the Langmuir and Freundlich models were tabulated in Table 2. It revealed that the correlation coefficients (R^2^) of the Langmuir fit curve (0.9851) were larger than that of Freundlich fit curve (0.8717), demonstrating that the adsorption of AC on N-SC/β-CD-MOFs conformed to Langmuir isotherm model. In other words, there was uniform monolayer adsorption on the surface of N-SC/β-CD-MOFs. The maximum adsorption capacity was calculated as 66.43 mg/g according to the Langmuir isotherm equations. Subsequently, N-SC/β-CD-MOFs were compared with other reported AC adsorbents (Appendix A). The result showed that *q_m_* from N-SC/β-CD-MOFs was higher than those from other reported adsorbents. Additionally, the equilibrium time of N-SC/β-CD-MOFs was only 30 min, which was significantly shorter than that of other listed adsorbents. All these demonstrated that N-SC/β-CD-MOFs were promising in removing AC from an aqueous solution.

#### 2.2.4. Adsorption Mechanism

In order to explore the functional groups on the surface of N-SC/β-CD-MOFs related to AC adsorption, FTIR spectra of N-SC/β-CD-MOFs before and after AC adsorption were compared (Appendix A). The result showed that the bands in 3437 cm^−1^ (−OH groups in phenol, carboxyl, or absorbed water), 1635 cm^−1^ (C=O groups in carboxyl and lactone groups), and 1220 cm^−1^ (C-O groups in ether or lactone) became weak and even disappeared after adsorption, demonstrating that oxygen-containing functional groups played an irreplaceable role in the adsorption process of AC through n–π interaction and hydrogen bond. This result was consistent with previous reports that the C=O group was generally considered to be a strong active group for adsorbing aromatic pollutants in solution [37].

#### 2.2.5. Desorption Studies

The result of desorption efficiency (*η_Des_*) by the chemical method is shown in Table 3. The best desorption efficiency was found when using ethanol (63.1%) and methanol (58.7%) as desorbing agents. In addition, NaOH (0.2 M) can desorb around 29.3% of AC from laden N-SC/β-CD-MOFs. The AC adsorbed by N-SC/β-CD-MOFs can be desorbed by HCl (0.2 M) and NaCl (0.2 M), with *η_Des_* being 9.72% and 5.33%, respectively. The results indicated the important contribution of surface interaction (i.e., hydrogen bonding and n–π interaction) during the adsorption process.

### 2.3. Electrochemical Detection of AC Based on N-SC/β-CD-MOFs/GCE

#### 2.3.1. Electrochemical Characterization of Various Modified Electrodes

Electrochemical impedance spectroscopy (EIS) was employed to explore the electron transfer performance of different modified electrodes (Appendix A). The Nyquist plots of bare GCE (curve a) and β-CD-MOFs/GCE (curve b) showed that the charge transfer resistances (R_ct_) of β-CD-MOFs/GCE (702.9 Ω) were clearly larger than that of bare GCE (449.5 Ω), mainly due to the inherent poor conductivity of MOFs. The R_ct_ of N-SC/GCE (53.3 Ω) was obviously lower than that of SC/GCE (232.5 Ω). Both the R_ct_ values of SC/GCE and N-SC/GCE were lower than bare GCE (449.5 Ω), revealing that SC possessed good conductivity, and N doping can improve the conductivity of SC. Furthermore, it was worth noting that the R_ct_ value of N-SC/β-CD-MOFs/GCE (168.3 Ω) was significantly lower than that of β-CD-MOFs, indicating the good conductivity of the composite. 

CVs of N-SC/β-CD-MOFs/GCE under different scan rates were investigated in Appendix A. It showed that the redox peak current of [Fe(CN)_6_]^3−^/^4−^ increased gradually with the increase in the scan rate. The good linear relationship (R^2^ = 0.99) between the square root of scan rate and the redox peak current (Appendix A) indicated that the redox of [Fe(CN)_6_]^3−^/^4−^ on N-SC/β-CD-MOFs/GCE belonged to a typical diffusion-controlled process. Then, the electrochemical effective surface area (A) of N-SC/β-CD-MOFs/GCE was calculated as 5.23 cm^2^ according to the Randles–Sevcik equation (Equation (5)) [38]. The N-SC/β-CD-MOFs with a large effective area could provide more active sites for electrochemical reactions, which would endow N-SC/β-CD-MOFs with excellent electrochemical catalytic performance:*I_p_* = 2.69 × 10^5^*AD*^1/2^*n*^3/2^*v*^1/2^*c*(5)
where *A* represents the effective electroactive surface area of N-SC/β-CD-MOFs/GCE; *v* represents the scan rate and *c* was the concentration of [Fe(CN)_6_]^3−/4−^ (5 mM); *n* (*n* = 1) is the electron transfer number; and *D* stands for the diffusion coefficient.

#### 2.3.2. Electrochemical Behaviors of AC on Various Electrodes

The electrochemical behaviors of 20.0 μM AC on various electrodes were studied by DPV (Appendix A). There was no obvious current response on bare GCE (curve a) and β-CD-MOFs/GCE (curve b). This might be caused by the poor conductivity of β-CD-MOFs. However, the SC/GCE (curve c) showed an obvious oxidation peak. On N-SC/GCE (curve d), the current signal was significantly increased, which could be attributed to the fact that doping with nitrogen endowed N-SC with excellent conductivity and electrocatalytic performance. Furthermore, it was worth noting that N-SC/β-CD-MOFs/GCE (curve e) exhibited the largest oxidation peak current. This phenomenon is because of the synergistic effect between β-CD-MOFs and N-SC. The β-CD-MOFs could facilitate the preconcentration of AC on the surface of N-SC/β-CD-MOFs/GCE via the host-guest recognition ability of β-CD and N-SC, with excellent conductivity and electrocatalytic performance, would catalyze the oxidation of AC. These results proved that N-SC/β-CD-MOFs owned outstanding catalytic performance and could be used for the construction of electrochemical sensors for AC.

#### 2.3.3. Optimization of Experiment Parameters

Some experiment parameters were optimized to get the best sensing performance. First, CVs of N-SC/β-CD-MOFs/GCE in 0.1 M PBS with different pH (4.0–9.0) containing 20.0 μM AC were researched (Figure 5A). The oxidation peak of AC increased with the increase in pH from 4.0 to 7.0, and then gradually decreased when the pH was beyond 7.0 (Figure 5B, red line). So, 7.0 was elected as the optimal pH for the next experiments. In addition, the oxidation peak potential of AC shifted negatively with the increase in pH value, indicating that protons were involved in the electrochemical oxidation process. The oxidation peak potential of AC showed a good linear relationship (R^2^ = 0.995, E_pa_ = 0.807–0.057 pH) with pH (Figure 5B, black line). The slope (0.057 V·pH^−1^) was closed to that of the Nernstian value (0.059 V·pH^−1^), indicating that the number of electrons and protons were equal in the electrocatalytic oxidation process of AC on N-SC/β-CD-MOFs/GCE [39].

The influence of proportions between N-SC and β-CD-MOFs on the catalytic current of AC was also explored (Figure 5C). As can be seen, the peak current of AC increased with increasing the ratio of N-SC and β-CD-MOFs from 1:0.25 to 1:1. This could be ascribed to the fact that more AC was enriched on the electrode surface with the increase in β-CD-MOFs, and then the catalytic current was amplified. However, the peak current decreased significantly with further increasing the content of β-CD-MOFs. It could be because excessive β-CD-MOFs with poor conductivity would hinder electron transfer on the modified electrode. Thus, 1:1 was selected as the optimal proportion of N-SC and β-CD-MOFs for subsequent experiments.

Finally, the influence of accumulation time on the electrochemical signal of AC was investigated (Figure 5D). It showed that the peak current of AC increased rapidly with the increase in accumulation time from 0 to 180 s and then basically stabilized. The result showed that the adsorption of AC on the surface of N-SC/β-CD-MOFs/GCE tended to be saturated when the adsorption time was 180 s. Thus, 180 s was preferentially utilized for the next experiments.

#### 2.3.4. Electrocatalytic Mechanism of AC on N-SC/β-CD-MOFs/GCE

The CV curves of N-SC/β-CD-MOFs/GCE under various scan rates in 0.1 M PBS (pH = 7.0) containing 20 μM AC were shown in Figure 6A. The peak currents of AC increased linearly with the scan rate from 10 to 200 mV s^−1^. Besides, Figure 6B presented a fine linear relationship (R^2^ = 0.99, I_pa_ = 0.408 *v* + 5.111 and I_pc_ = −0.393 *v* − 4.272) between peak currents (I_pa_ and I_pc_) and scan rate (*v*), manifesting that the electrochemical process on N-SC/β-CD-MOFs/GCE belonged to a surface adsorption-controlled process.

Additionally, it can also be observed that with the increase in scan rate, the oxidation peak potential (E_pa_) of AC shifted positive and the reduction peak potential (E_pc_) of AC shifted negative gradually. A positive correlation was found between peak potential and ln *v* under high scan rates (Figure 6C). Their linear regression equations could be expressed as E_pa_ = 0.247 + 0.037 ln *ν* and E_pc_ = 0.453 − 0.031 ln *ν*, respectively. Then, according to Laviron’s theory (Equations (6) and (7)), the electron transfer coefficient (α) and electron transfer number (n) in the redox process of AC were calculated as 0.5 and 2, respectively, indicating there was a two-electron and two-proton transfer process in the redox process of AC on N-SC/β-CD-MOFs/GCE [40].
E_pa_ = E^0^ + (RT/αnF)ln(RTk^0^/αnF) + (RT/αnF)ln*v*
(6)
E_pc_ = E^0^ + {RT/(1 − α)nF}ln{RTk^0^/(1 − α)nF} − {RT/(1 − α)nF}ln*v*
(7)

#### 2.3.5. Electrochemical Detection of AC Based on N-SC/β-CD-MOFs/GCE

Electrochemical sensing performance of AC based on N-SC/β-CD-MOFs/GCE was performed by DPV under the optimal conditions. The result displayed that the oxidation peak current of AC continued to increase with the increase in the concentration (Figure 7A), and there was a good linear relationship in the range of 1.0–30.0 μM (Figure 7B). The LOD was calculated to be 0.3 nM (S/N = 3). The performance of N-SC/β-CD-MOFs /GCE was compared with other AC sensors (Appendix A). It was evident that the sensor proposed in this work displayed obvious advantages in both linear range and detection limit, which were associated with the excellent conductivity, abundance of active sites, and large surface area of N-SC/β-CD-MOFs. These results indicated that N-SC/β-CD-MOFs were promising electrode modified materials for sensitive detection of AC.

#### 2.3.6. Reproducibility, Repeatability, Stability and Anti-Interference of N-SC/β-CD-MOFs/GCE

The reproductivity of N-SC/β-CD-MOFs/GCE was investigated by detecting AC with five randomly selected electrodes under the same conditions. The relative standard deviation (RSD) of the results detected by five electrodes was 2.83% (Figure 8A), indicating that the reproducibility of N-SC/β-CD-MOFs/GCE was wonderful.

Subsequently, the repeatability of the sensing platform was investigated using one N-SC/β-CD-MOFs/GCE to continuously examine 15 times (Figure 8B). The satisfactory RSD (2.63%) illustrated the good repeatability of N-SC/β-CD-MOFs/GCE. Moreover, N-SC/β-CD-MOFs/GCE was employed to detect 20.0 μM AC every two days to investigate the stability. As a result, the response current of the AC retained 96.9% after 20 days (Figure 8C), confirming that the N-SC/β-CD-MOFs/GCE exhibited great stability.

The anti-interfering ability of N-SC/β-CD-MOFs/GCE was researched in the presence of several potential interferents. The result showed that there were no significant changes for the peak current of 20.0 μM AC in the presence of a 100-fold concentration of K^+^, Na^+^, NO_2_^−^, Cl^−^, and 10-fold concentration of ascorbic acid, glucose, ibuprofen, and diclofenac (Figure 8D). These results demonstrated that N-SC/β-CD-MOFs/GCE owned excellent anti-interference ability for the detection of AC.

#### 2.3.7. Real Samples Analysis

In order to prove the practical application of the proposed sensor, N-SC/β-CD-MOFs/GCE was employed to detect AC in the lake water by DPV. Before detection, lake water was pretreated according to the method in Section 3.9. Then, AC with specific concentrations (0, 1.0, 3.0, 5.0, and 10.0 μM) was added separately to 10 mL of treated lake water. The concentration of AC in those real samples detected by N-SC/β-CD-MOFs/GCE was listed in Appendix A. The excellent recovery (95.0–104.2%) and RSD (2.27–3.25%) confirmed that the proposed sensor was potentially applicable in the real sample.

## 3. Experimental

### 3.1. Reagents

The reagents, as well as their purity, were displayed in the Appendix A.

### 3.2. Apparatuses

The apparatuses used in this work were provided in the Appendix A.

### 3.3. Preparation of N-SC

Soulangeana sepals were first washed with ultrapure water, followed by drying and grinding into powder. Then, 1 g of powder was immersed in 10 mL of KOH solution (20 wt%) and stirred for 24 h. Subsequently, the powder was collected by centrifugation and carbonized at 800 °C under an Ar atmosphere for 120 min. Finally, the carbonized product was washed with 1 M HCl and ultrapure water, which was then filtrated, vacuum dried, and named SC. For comparison, the powder without the KOH activation process was also carbonized under the same conditions and named NKSC.

Melamine (2 g) and SC (0.4 g) were dispersed in 400 mL of ultrapure water and stirred continuously for 48 h. Subsequently, the precipitate was dried at 60 °C. Finally, the obtained powder was calcined at 900 °C for 1 h under an argon atmosphere to obtain N-SC.

### 3.4. Preparation of β-CD-MOFs

Firstly, β-CD (1.13 g) and KOH (0.45 g) were dissolved in a small beaker containing 20 mL ultrapure water. Then, the beaker was placed in a large beaker containing 150 mL of methanol. In there, methanol was evaporated slowly and diffused into the aqueous solution. One week later, the white precipitate in a small beaker was collected and washed with methanol several times. Later, the white precipitate was soaked in dichloromethane for 72 h. Finally, the precipitate was obtained after drying in the oven at 60 °C for 10 h, which was named as β-CD-MOFs.

### 3.5. Preparation of N-SC/β-CD-MOFs

The preparation of N-SC/β-CD-MOFs composite was carried out using bath sonicate. Specifically, N-SC and β-CD-MOFs were dispersed in the water at the weight ratio of 1:1 and sonicated under 60 W for 60 min. Then, the products were collected by centrifugation and dried under vacuum to obtain the N-SC/β-CD-MOFs composite.

### 3.6. Adsorption Studies of AC

All adsorption experiments were performed in a 50 mL beaker. A known amount of N-SC/β-CD-MOFs (0.005–0.065 g) was added to 26 mL of AC solution, with the concentration ranging from 5 to 300 mg·L^−1^. 0.1 M HCl and NaOH solutions were used to adjust the pH value of the AC solution, ranging from 3 to 11. All adsorption experiments were performed at room temperature under continuous stirring with a magnetic stirrer. After adsorption, the suspension was filtered through a 0.22 μm filter membrane, and the concentration of AC was determined by a UV-1800 spectrophotometer at 241 nm. The adsorption capacity (q, mg·g^−1^) and removal efficiency (R, %) of AC were calculated according to Equation (8) and (9), respectively:
(8)
$$q=\frac{\left(C_{0}-C_{e}\right)V}{m}$$


(9)
$$R=\frac{\left(C_{0}-C_{e}\right)}{C_{0}}\times 100\%$$

where *q* (mg/g) is the adsorption capacity of N-SC/β-CD-MOFs toward AC; *c*_0_ (mg/L) and *c_e_* (mg/L) are the initial concentration and the equalized concentration of AC, respectively; *V* (L) is the volume of AC solution; *m* (g) is the weight of N-SC/β-CD-MOFs; and *R* (%) is the removal efficiency of AC.

### 3.7. Desorption Studies of AC

Desorption experiments were carried out using: the laden N-SC/β-CD-MOFs were washed with water to remove any unadsorbed AC and dried overnight in an oven at 50 °C.

Subsequently, the laden N-SC/β-CD-MOFs were desorbed by different desorbing agents, including HCl (0.2 M), NaOH (0.2 M), NaCl (0.2 M), ethanol, and methanol. The solid/liquid ratio for the desorption study was maintained at 0.77 g/L. After shaking at 150 rpm for 6 h, the residual concentration of AC in the solution was determined. The percentage of AC desorbed (*η_Des_*) from laden N-SC/β-CD-MOFs was calculated from Equation (10):
(10)
$$\eta_{Des}=\left(1-\frac{q_{e}-q_{d}}{q_{e}}\right)\times 100\%$$

where *q_e_* (mg/g) represents the equilibrium adsorption capacity of AC and *q_d_* (mg/g) is the amount of AC desorbed from the laden N-SC/β-CD-MOFs.

### 3.8. Preparation of Modified Electrodes

Firstly, bare GCE was polished with alumina slurry and ultrasonically washed with ultrapure water and ethanol. Then, 5 μL of N-SC/β-CD-MOFs dispersion (5 mg/mL) was pipetted on the surface of polished GCE to obtain N-SC/β-CD-MOFs/GCE. SC/GCE, N-SC/GCE, and β-CD-MOFs/GCE were prepared according to the same way, except that N-SC/β-CD-MOFs were replaced by SC, N-SC, and β-CD-MOFs.

### 3.9. Preparation of Real Samples

Lake water samples were collected from Cui Lake at Jiangxi Agricultural University. The collected lake water was filtered through a 0.45 μm filter. Then, the filtered lake water was diluted 100-fold with PBS (0.1 M, pH = 7.0), which was used as real samples and stored in a refrigerator at 4 °C for later use. Finally, the standard addition method was used to research the detection performance of N-SC/β-CD-MOFs/GCE toward AC in real water samples.

## 4. Conclusions

In this study, the N-SC/β-CD-MOFs composite was successfully prepared for sensitive detection and efficient removal of AC in water. As a result of the porous structure and host-guest recognition ability, N-SC/β-CD-MOFs displayed a good adsorption effect on AC, and the maximum adsorption capacity of N-SC/β-CD-MOFs toward AC was as high as 66.43 mg/g. The pseudo-second-order kinetics model and the Langmuir isotherm model best represented the adsorption between N-SC/β-CD-MOFs and AC, indicating that the adsorption was mainly chemical adsorption. Meanwhile, as sensing materials, N-SC/β-CD-MOFs showed a high catalytic effect on the oxidation of AC with a low LOD (0.3 nM), wide linear range (1.0–30.0 μM), as well as satisfactory reproducibility and anti-interference. In addition, it displayed good application prospects for detecting AC in real samples. In a word, this work integrates the functions of efficient removal and sensitive electrochemical detection, which provides a new idea for the simultaneous detection and removal of pollutants in water.

## Data Availability

The data presented in this study are available in article.

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
