# Peer review of "Nitrogen Doped Porous Biochar/β-CD-MOFs Heterostructures: Bi-Functional Material for Highly Sensitive Electrochemical Detection and Removal of Acetaminophen"

_molecules, 2023, doi:10.3390/molecules28062437_

Round 1
Reviewer 1 Report
Comments and Suggestions for Authors
Wang and coworkers report the synthesis of Nitrogen Doped Porous Biochar/beta-CD-MOFs by mixing the as-synthesized N-SC and beta-CD-MOF under a simple condition. The as-prepared hybrid material was characterized by PXRD, SEM, XPS, suggesting the successful formation of the material. The resulting material was further utilized in sensing and enrichment of AC, and it showed a promising result with high sensitivity, superior adsorption capacity and recyclability. In general, given the structural novelty, the scope of this work will be well perceived by the community of MOFs or sensing, and attract general readership of Molecules. Therefore, I am happy to recommend publication, but only after the following issues are addressed.
1. The manuscript was written in poor English manner with too many mistakes. It should be largely improved and carefully proofread before the publication.
2. In the figure 3b, author compared two types models to fit the adsorption curve. Apparently, pseudo first-order model doesn’t fit with the data. However, in the table 1, a high R2 value of 0.98 was calculated. That seems strange.
3. Author may want to add the PXRD or SEM characterization of the material after the adsorption cycles.
4. Not sure if author mentioned that whether the adsorbed AC could be desorbed from N-SC/β-CD-MOFs or not.
Reviewer 2 Report
Comments and Suggestions for Authors
Journal: Molecules
Ms. ID.: molecules-2196222
Title: Nitrogen Doped Porous Biochar/β-CD-MOFs Heterostructures for Highly Sensitive Electrochemical Detection and Removal of Acetaminophen
Yu et al. proposed nitrogen-doped Soulangeana sepals derived biochar/β-cyclodextrin-Metal- organic frameworks (N-SC/β-CD-MOFs) composite for simultaneous efficient removal and detection of acetaminophen. The composite was synthesized, characterized, and used as an adsorbent for acetaminophen removal. Moreover, the composite was used for the electrochemical detection of acetaminophen. The manuscript is very interesting. It is very well prepared and fits well with the scope of the Journal.
The abstract is suitable.
The introduction is informative and addresses all necessary aspects.
The results are well presented. The Discussion is carefully composed and conducted with respect to the existing literature.
The experimental section is detailed enough and the methods used are adequately described.
The conclusions are in line with the presented results.
I consider the manuscript suitable for publication after the authors address the following issue. It would be essential to check if there is any hydrolysis during the adsorption process. Since the acetaminophen concentration is determined by UV-Vis, you cannot be sure that there are no new products during this process. HPLC would help resolve this issue.
Round 2
Reviewer 1 Report
Comments and Suggestions for Authors
The reviewer is satisfied with the current version and it can be published.